# Perspectives of Statistician, Microbiologist, and Clinician Stakeholders on the Use of Microbiological Outcomes in Randomised Trials of Antimicrobial Stewardship Interventions

**DOI:** 10.3390/antibiotics12050885

**Published:** 2023-05-10

**Authors:** Tin Man Mandy Lau, Rhian Daniel, Kerenza Hood, Mandy Wootton, Kathryn Hughes, Beth Stuart, Gail Hayward, Tamas Szakmany, David Gillespie

**Affiliations:** 1Centre for Trials Research, Cardiff University, Cardiff CF14 4YS, UK; hoodk1@cardiff.ac.uk (K.H.); gillespied1@cardiff.ac.uk (D.G.); 2Division of Population Medicine, Cardiff University, Cardiff CF14 4YS, UK; danielr8@cardiff.ac.uk; 3Specialist Antimicrobial Chemotherapy Unit, University Hospital of Wales, Public Health Wales, Cardiff CF14 4XW, UK; mandy.wootton@wales.nhs.uk; 4PRIME Centre Wales, Division of Population Medicine, Cardiff University, Cardiff CF14 4YS, UK; hugheska6@cardiff.ac.uk; 5Pragmatic Clinical Trials Unit, Queen Mary University of London, London E1 2AB, UK; b.l.stuart@qmul.ac.uk; 6Nuffield Department of Primary Care Health Sciences, University of Oxford, Oxford OX2 6GG, UK; gail.hayward@phc.ox.ac.uk; 7Critical Care Directorate, Grange University Hospital, Aneurin Bevan University Health Board, Cwmbran NP44 8YN, UK; szakmanyt1@cardiff.ac.uk; 8Department of Anaesthesia, Intensive Care and Pain Medicine, Cardiff University, Cardiff CF14 4XN, UK

**Keywords:** antimicrobial resistance, microbiological outcome, statistical approaches, stakeholder engagement

## Abstract

Microbiological data are used as indicators of infection, for diagnosis, and the identification of antimicrobial resistance in trials of antimicrobial stewardship interventions. However, several problems have been identified in a recently conducted systematic review (e.g., inconsistency in reporting and oversimplified outcomes), which motivates the need to understand and improve the use of these data including analysis and reporting. We engaged key stakeholders including statisticians, clinicians from both primary and secondary care, and microbiologists. Discussions included issues identified in the systematic review and questions about the value of using microbiological data in clinical trials, perspectives on current microbiological outcomes reported in trials, and alternative statistical approaches to analyse these data. Various factors (such as unclear sample collection process, dichotomising or categorising complex microbiological data, and unclear methods of handling missing data) were identified that contributed to the low quality of the microbiological outcomes and the analysis of these outcomes in trials. Whilst not all of these factors would be easy to overcome, there is room for improvement and a need to encourage researchers to understand the impact of misusing these data. This paper discusses the experience and challenges of using microbiological outcomes in clinical trials.

## 1. Introduction

Microbiological outcomes, including the incidence or prevalence of organisms resistant to a certain antimicrobial, the prevalence of multidrug-resistant organisms, and the incidence of infections due to specified organisms, are essential in measuring the effectiveness of strategies aiming to control the growth of antimicrobial resistance (AMR), as well as being used to diagnose or define infection [1,2]. Advice on how to use microbiological outcomes in antimicrobial stewardship interventions (ASIs) is covered in several official guidelines [1,2,3,4,5].

Recently, we conducted a systematic review to investigate the use of microbiological outcomes in randomised controlled trials (RCTs) evaluating ASIs [6]. A narrative synthesis approach was taken to summarise the quality of reporting on microbiological outcomes in ASIs. Several issues were identified including: (i) relatively few trials of ASIs (15.4%) reported microbiological outcomes, (ii) details about sample collection and reasons for missing samples were poorly reported, (iii) laboratory procedures for sample processing and guidelines used to define an infection and resistance were inadequately detailed, (vi) the selected study population for analysis was frequently based on all randomised participants without consideration of whether the sample data were obtained, and (v) microbiological outcome data were typically operationalised as dichotomous outcomes.

The findings from this work suggested the following: (i) there is a lack of high-quality evidence around the microbiological impacts of ASIs; (ii) there is potential bias if samples are differentially missing by study sites, participant characteristics or trial arms; (iii) quality assurance is needed to ensure the laboratory works are transparent; (vi) outcomes can be misleading (true negative and missing samples unreported). Additionally, microbiological data are complex and multi-dimensional, and using dichotomous outcomes (often based on a composite of several variables) may be inadequate, compared to an operationalisation that could more appropriately respect the complex and high-dimensional structure inherent in these data.

To understand the challenges of using microbiological data and to improve the use of these data, a stakeholder engagement group was convened. This paper has been developed with the input of the individuals from a stakeholder engagement group including their perspectives on the use of microbiological outcomes in clinical trials, challenges and potential solutions in handling and reporting microbiological data, and the identification of statistical methods and strategies to maximise the use of microbiological data.

## 2. Results

### 2.1. Perspectives on the Issues Identified from the Systematic Review Work

Given the stakeholders’ roles and experiences in clinical trials, none were surprised by the issues identified from the systematic review. The phenomenon of dichotomising or categorising microbiological outcomes was a long-recognised issue. In particular, the statisticians described how the limited scope of trial funding, planning, and setting (primary/secondary care) of the trial caused problems in using microbiological data and therefore microbiological outcomes were often not considered as part of trial outcome packages. Funding for clinical trial methodology research in this area was thought to help improve the quality of clinical research findings. However, given the lack of funding for methodological work in ASIs, there is insufficient time or capacity for statisticians to engage in high-dimensional microbiological analysis, and hence microbiological data are dichotomised or categorised during the analysis even though this oversimplifies the data and may not be appropriate. It was felt that research questions related to resistance should be more clinically relevant and better thought out. Therefore, researchers need to define trial objectives, microbiological outcomes, and estimands with greater care involving in-depth discussion and understanding among microbiology, clinical, and statistical experts.

The microbiologists felt that microbiological data have been misused and misunderstood in research and they pointed to several key issues: first, microbiological data are often kept in the background due to their complexity; second, there are no fixed definitions for microbiological outcomes across studies; and third, researchers lack knowledge in the processes used by laboratories to analyse the samples. The microbiologists also pointed out that the interpretation of microbiology results varied between UK laboratories. This is because there is often room for varied interpretation in the UK Standards for Microbiology Investigations, which are the standard operating procedures produced by the government [7]. Moreover, it is unclear how clinically meaningful these dichotomous variables are and how these outcomes support trial findings. Hence, some explanations of the microbiological outcomes should be included in the publication. The method of dichotomisation should be carefully considered as a dichotomous microbiological outcome could lead to misclassification bias if there are high variations between laboratories. This risk needs to be balanced against the potential for variation in practices and across laboratories and awareness that anything at a more granular level could introduce considerable noise into any resulting analysis.

### 2.2. Perspectives on the Value of Using Microbiological Data in Trials of ASI

The discussions around the value of using microbiological data can be roughly divided into four areas: the relationship between antibiotic use and AMR, whether microbiological outcomes should be considered as primary or secondary study outcomes, the value of using microbiological data, and the patient’s perspective on AMR.

There was general agreement that although the relationship between antibiotic use and AMR is well recognised, its extent and form are not fully understood (i.e., AMR can be developed and spread, from patient to patient in health care facilities, from contaminated water and soil, or transmitted between animals and humans and between bacteria on plasmids). The situation differs from, say, cigarette use and lung cancer, where the relationship is so unequivocal that the use of smoking cessation, say, as a primary outcome is acceptable. Therefore, a microbiologically confirmed AMR outcome should ideally be collected in the trial of ASIs instead of antibiotic usage data only.

There were different views on including microbiological outcomes as the primary outcome in trials of ASIs. As ASIs normally result in the restricted use of antimicrobials (compared to their usual level of use), ASIs are usually designed with clinical outcomes assessed for non-inferiority, with more recent movements towards co-primary outcomes investigating clinical outcomes (non-inferiority framework) and antimicrobial use (superiority framework) [8]. Replacing the antimicrobial use co-primary outcome with a microbiological co-primary outcome (e.g., an outcome capturing AMR), it was thought to be difficult to recruit an individual into a trial where the intervention may be associated with slightly worse clinical outcomes even if lower levels of AMR could be achieved. This point was agreed upon by stakeholders from the primary care setting. In contrast, stakeholders from the secondary care setting had a different view and suggested that microbiological outcomes can be potentially considered as a primary outcome in the hospital setting for interventions that change the practice or in the treatment of or handling of patients with infectious disease. Overall, stakeholders agreed that including microbiological outcomes in ASIs would benefit patients and support the trial findings.

Another critical point highlighted by the microbiologists was that a lack of sufficient communication between microbiologists and infectious disease doctors when designing and conducting these trials meant that microbiological outcomes were considered less important than other outcomes (e.g., clinical outcomes). Therefore, multiple separate views on the importance of microbiological outcomes in trials were found.

According to their experience, stakeholders had different perspectives on whether patients had an awareness of and viewed AMR as important. Stakeholders felt that this would depend on the patient’s own experience; for example, patients with long-term urinary tract infections who have been prescribed long-term antibiotics may be more aware of AMR while otherwise healthy individuals may be less likely to consider the harmfulness of AMR when prescribed antibiotics in general practice.

### 2.3. Perspectives on the Current Microbiological Outcome

According to the findings from the systematic review, the creation of composite measures represented as dichotomous variables was the most common method used to summarise microbiological outcomes, such as the presence or absence of infection (e.g., microbiologically defined *Clostridioides difficile* infection) or the presence of resistance. These composite measures can also take on more complex forms (e.g., microbiologically defined UTI can encompass the growth of certain bacteria at certain levels with the presence of a certain quantity of white blood cells). Although these approaches reduce the time taken for analysis and enable quicker decision making for clinicians, this method comes with several disadvantages, which include information loss, reduced statistical power, and underestimated variation in outcomes between groups [9].

Based on the stakeholders’ experience and knowledge, microbiological datasets frequently contained missing data. The potential of combining and synthesising similar sub-sets from different trials was suggested; for example, an analysis of those who have a valid sample, or combining participants who have microbiological confirmed urinary tract infections from several trials. The challenge here is that there are variations between laboratory procedures and the microbiological data collected from each study could differ. This also could bring two additional challenges: first, managing measurement error, and second, contamination. The microbiologists also pointed out that inappropriate sample collection is another well-known issue, and guidance on how to collect samples that minimise errors and contamination is needed.

Additionally, other more clinically relevant microbiological outcomes should be considered in ASIs, including how long the participant carried AMR bacteria, and whether the resistant organisms were transmitted to other participants or whether the resistance genes transferred to other bacteria, including commensals. Moreover, the microbiological outcome can act as an intermediate outcome between the intervention and the clinical outcomes, which is another important element to investigate [10]. Although the discussions were mainly around AMR, the stakeholders agreed that the whole microbiome system should be considered rather than focusing only on the resistance element.

### 2.4. Perspectives on the Alternative Statistical Approaches

Two questions were raised around the rationale for alternative statistical approaches, including whether we are looking for a better binary/categorical outcome, standardising microbiological outcomes/statistical methods for handling microbiological outcome data, or whether we would like to estimate the probability that a particular organism is resistant to a particular antibiotic. In response to these questions, the statisticians described the intention of using more sophisticated statistical methods aimed to improve the quality of the microbiological outcomes by allowing this complex relationship rather than summarising the data into one-dimensional (binary) variables. The end goal of the alternative statistical approach is to answer the research question well.

Three statistical methods were mentioned and supported by the stakeholders, including principal component analysis, latent variable modelling, and structural equation modelling. Principal component analysis can be used as part of the descriptive analysis to reduce the dimensionality and increase the interpretability of the microbiology data. The idea of latent variable modelling is to relate a set of microbiological variables (observable variables) to create a set of indicators (latent variables), for example, a high-risk patient indicator and a health indicator. Structural equation modelling is an approach in which path-specific effects between observed (and sometimes latent) variables are estimated, given a set of statistical and structural (graphical) assumptions.

However, the generalisability of latent variable modelling was questioned due to this approach being data-driven, and therefore not directly replicable in a different study. Microbiological data will differ across settings and populations. For example, the amount of AMR is likely to depend on the antimicrobial the population has been exposed to and the different environments; hence, the output of the latent variable modelling would differ between studies in different clinician sites and populations. However, the idea of performing the latent variable modelling is not to generalise a single standardised set of classes and replicate these in other studies, but instead to answer the research questions appropriately and potentially guide trial teams towards using high-dimensional analytical approaches with greater standardisation of condition-specific indicator variables.

### 2.5. Other Challenges

However, during the discussions, several additional barriers to including microbiological outcomes in clinical research were raised. These barriers were outside of the four predefined discussion points and suggest that more work is needed in this field to improve the quality of microbiological outcomes:Microbiological outcomes are frequently excluded in publications as it was felt that journals were not interested.If microbiological outcomes are reported in publications, it was felt that journals were not interested in detailed microbiological outcomes.The collection of samples for research is difficult in routine care and collecting samples under the gold standard (e.g., using a method that minimises errors and reduces the risk of contamination) for large-scale trials is even more challenging and can be prohibitively expensive.Work is needed to understand the mechanistic work in the laboratory and AMR.Related to the above, study funding for mechanistic work was mentioned by two members of the clinician group. The funding issues were similar to the point made by the statisticians earlier during the meeting; there was not enough or no funding available to conduct the mechanistic work or the analysis.Database development was felt to be another important issue as microbiological data are often complex and in a multi-layer structure.

## 3. Method

A stakeholder mapping exercise was conducted to identify potential stakeholders. The mapping exercise laid out the candidate stakeholders in terms of their experience, primary research interest, and the types of input and influence they required. Following this exercise, researchers with knowledge of AMR, complex data analysis, and experience in conducting clinical trials (especially RCTs) were the key stakeholders approached.

A total of ten researchers formed the stakeholder engagement group including three clinical microbiologists, three statisticians (two medical statisticians and a biostatistician), and four clinicians (two primary-care and two secondary-care clinicians). These stakeholders were either approached individually or from a European network (General practice Research on Infection Network). All stakeholders were checked for eligibilities (i.e., did they have experience with the RCT design, conduct, analysis of trials of ASIs, and were they either a statistician, clinician, or microbiologist). This single meeting was conducted online and lasted for two hours. The discussions for the stakeholder engagement group were based on four predefined discussion points. Any additional challenges were outlined following this. There was no financial compensation for the participants.

## 4. Conclusions

In this paper, the stakeholders’ perspectives on the challenges in handling, analysing, and reporting microbiological data are described and summarised in Figure 1. Not all of these challenges are easy to overcome, and further evidence around the value (e.g., clinical and microbiological relevance and statistical properties) of alternative statistical methods that make better use of complex microbiological data is needed to improve the quality and value of microbiological data in trials of ASIs. The limitation of this current work is a lack of patient perspective about the measurement and analysis of AMR in trials of ASI. Guidelines on how to handle and report these microbiological outcomes are needed. Extra attention is required from journals, researchers, and funders when considering these outcomes. Here, we highlight two future steps. These are:Additional efforts from statisticians are needed in terms of the analyses and publications to minimise the risk of missing data or unnecessary results, including applying more sophisticated methods suggested by the stakeholders to appropriately analyse microbiological outcome data.Guidelines should be developed, including a list of critical elements when using, reporting, and analysing microbiological outcomes in clinical trials of ASIs.

## Figures and Tables

**Figure 1 antibiotics-12-00885-f001:**
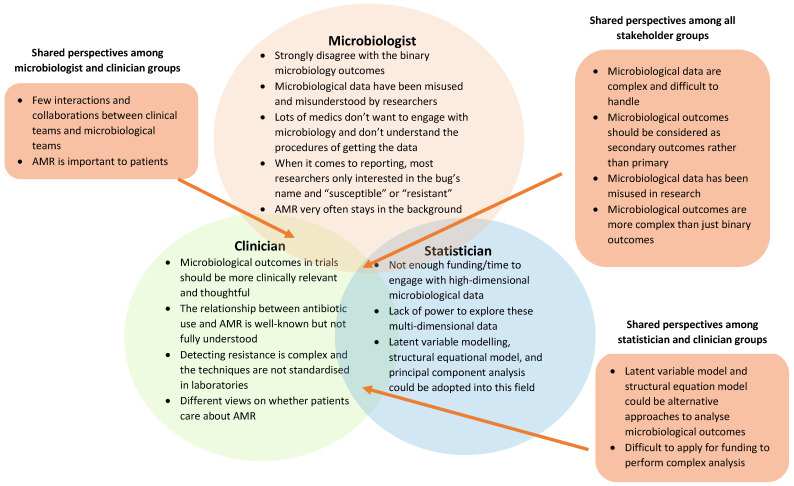
Shared and unique perspectives among stakeholder group participants about the values and issues of handling microbiological data.

## Data Availability

Not applicable.

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
