# Peer review of "Perspectives of Statistician, Microbiologist, and Clinician Stakeholders on the Use of Microbiological Outcomes in Randomised Trials of Antimicrobial Stewardship Interventions"

_antibiotics, 2023, doi:10.3390/antibiotics12050885_

Round 1

Reviewer 1 Report

I found the manuscript interesting and considering important topics I have a few recommendations for authors.

1. How the authors decided to use exactly clinical microbiologists, statisticians, and clinicians? It sound logically, but the relation between abovementioned experts and those found in the systematic review (included publications) may be exist.

2. The number 3.4.i s twice used in the "Results" section and the name of subheadings 3.1. and 3.2 are absolutely the same. Renaming the subheadings will support the readers in text understanding.

The section 3.5. Other challenges are well explained, but reading the manuscript, I understand that need of additional education of some healthcare professionals exist. May be the development of guideline will reduce the gaps in this area, but additional efforts from statisticians in the way of presenting the data in publications could also minimize the risk of missing or unnecessary results.

Author Response

Thank you so much for reviewing this manuscript. We appreciated the time and effort that you and the reviewers have dedicated to providing your valuable feedback on the manuscript. We are delighted by the positive comments about our work and are pleased that we feel all of the comments can be addressed. We have highlighted the changes within the manuscript.

Here is a point-by-point response to the your comments and concerns.

Response 1: The focus for this stakeholder engagement group was randomised trials of antimicrobial stewardship interventions. In the method sections, we have mentioned that a stakeholder mapping exercise was conducted to identify potential stakeholders. All types of trialists were considered in the stakeholder mapping. The mapping exercise laid out the candidate stakeholders in terms of their experience, primary research interest, and the types of input and influence they required. Since the decision of using what type of outcomes is designed at the grant application stage and study development stage, we have selected trialists that will be involved at these stages and hence microbiologists, statisticians, and clinicians are selected. An eligibility check for each recruited stakeholder was conduced before the meeting to ensure they have the relevant experience and knowledge, this has been added in the method section.

Response 2: Thank you for spotting this. The first heading was an error and has now been corrected.

Response 3: Thank you for this, this comment is very helpful and we have added this under the conclusion section.

Reviewer 2 Report

The article “Perspectives of statistician, microbiologist, and clinician stakeholders on the use of microbiological outcomes in randomised trials of antimicrobial stewardship interventions” by Tin Man Mandy Lau et al. investigates the opinions of relevant stakeholders on microbiological outcomes in randomised trials of antimicrobial stewardship interventions, including current microbiological outcomes reported in trials, and alternative statistical approaches.

General remark:

The method of a focus group discussion with the included stakeholders is appropriate for the study aims. The manuscript is well written and structured. My major comment is that some statements are very generic and little specific, and therefore are little helpful for researchers who want to improve their ASIs studies. The authors should try to be more specific. Is it possible to come up with an example for a “perfect microbiological data set” and its analysis? In my opinion, the target group for this article are researchers who want to conduct (interventional) studies on the efficacy / effectiveness of ASIs. When describing the results of the stakeholder discussions, the authors should keep this target group in mind.

My other major comment is that this article depends on the results and conclusions from the mentioned systematic review and therefore, this manuscript feels like an extension of the systematic review rather than a standalone publication. I suggest that the authors revise the introduction so that readers who are not familiar with the systematic review can also follow this article (also see comments below).  

Specific comments:

- Introduction: I suggest that the authors briefly mention some typical examples of microbiological outcome data. For readers who are not familiar with the cited review article, this would help to follow this article.

- Methods: The authors recruited ten “key stakeholders”. In my opinion, it is important to characterize these stakeholders briefly in more detail: In which country / health system do they work? How much working experience do they have in their fields? If the key stakeholders were only from Wales / UK, the authors should discuss potential biases. Moreover, the authors should mention (i) how many meetings were conducted (presence or online meetings), (ii) how long were the meetings, and (iii) were there any financial compensations for the participants.

-  Results: “This section may be divided by subheadings. It should provide a concise and precise description of the experimental results, their interpretation, as well as the experimental conclusions that can be drawn.” This sentence probably should be removed.

- Results: “Funding often plays an important role to improve the quality of research.” What does this mean? Funding is the prerequisite of any research but its connection to the quality of research is not clear for me.

- Results: “Given the lack of funding for ASIs, there is insufficient time or capacity for statisticians to engage in high-dimensional microbiological analysis, and hence microbiological data are dichotomised or categorised during the analysis even though this oversimplifies the data and may not be appropriate.” I am not sure whether I understand this sentence. Why are ASIs not funded? In my experience third-party funders do not exclude proposals for studies investigating the efficacy of ASIs. Moreover, as I said above, an example of (dichotomized) microbiolgiocal outcome data” would be helpful.

- Results: “Therefore, researchers need to define trial objectives, microbiological outcomes, and estimands with greater care.” This statement is very generic. It is obvious that researchers always need to define trial objectives, microbiological outcomes, and estimands with greater care.

- Results: There is a contradiction: In line 127 the authors conclude that the relationship between antibiotic use and AMR is well-recognised, although its extent and form is not fully understand and few lines later the authors write “However, the relationship between antibiotic use and AMR is uncertain as this relationship is complex”.

- Results: “… that a lack of sufficient communication between microbiologists and infectious disease doctors meant that microbiological outcomes were considered less important than other outcomes (e.g. clinical outcomes).” The authors should explain this sentence. Who considers the microbiological outcomes less important than other outcomes? What do the authors mean with “lack of sufficient communication”? Do they mean the communication in the clinical routine or during the research study? In my personal opinion, I also support the notion that clinical (patient) outcomes are more important than other non-patient-related outcomes. In the end, the aim of clinical research is to improve the health of patients. On the other hand, I understand that it is difficult in clinical trials to show that effective ASIs translate into improved patient health outcomes (mortality, morbidity, health-related life quality).

- Results: Lines 156 – 161: This paragraph on the experience of the stakeholders on patient views on AMR does not add much on the aims of the study. The authors may consider removing this part.

- Results: Lines 183 – 186: Although these outcomes are very relevant, it is very challenging to measure these outcomes in a clinical study.

-  Results: Line 187: What is an ”intermediate outcome”?

-  Results: Section 3.4.: From my own research experience, I can confirm that in a perfect world with “perfect data”, better statistical methods can be used to obtain meaningful results and conclusions. However, a study is always a compromise between the collection of perfect and comprehensive data and the feasibility of the study. This is also true for all involved professions. Each profession has its own perfect and meaningful data. The challenge is to find outcome data (and ways to collect them!) that are accepted by all involved stakeholders and have scientific value to the study aims.

- Results: Section 3.5.: Point 1 and 2 are quite similar and could be merged

- Results: Lines 240 – 241: Why is database development important? Is it feasible to build databases for one clinical trial?

- Conclusion: Again, some statements are very generic and not helpful for readers. For example, “… further evidence is needed to improve the quality and value of microbiological data in trials of ASIs.”  What kind of evidence?

Author Response

Thank you so much for reviewing this manuscript. We appreciated the time and effort that you and the reviewers have dedicated to providing your valuable feedback on the manuscript. We are delighted by your comments about our work. We have highlighted the changes within the manuscript.

Here is a point-by-point response to the your comments and concerns.

Response to general remark: The purpose of this paper is to summarise the discussions during the stakeholder meeting around the issues of microbiological outcomes in randomised trials of ASIs. The statement made in this paper are the conversations from the stakeholders and therefore we aren't able to change their discussions or add information that did not happen during this meeting. Additionally, there is a limited word count within this brief report.
You are correct that this stakeholder group involved researchers who want to conduct (interventional) studies on the efficacy/effectiveness of ASIs.
This is a very interesting point about a “perfect microbiological dataset and its analysis”. However, we don’t think there is one for now but maybe one in the future but this is beyond the scope of this work.
We cannot see this as an issue of this paper being a standalone publication. Firstly, the systematic review has been published. Secondly, it is common to have a publication that is an extension of the previous research and we have ensure the reader know this in the introduction section. Thirdly, we confirmed that we have provided a reference for the systematic review in this paper. Since the research questions and objectives are different between the systematic review and the stakeholder group, we think this paper should be a standalone publication. We have modified some text so the reader is clear that the section is referred to the systematic review and the reader could look at the reference referred to if they are interested in the previous work.

Response 1: Thank you, this is has now being done (line 40-42) and the appropriate reference has also been added.

Response 2: Thank you for this we agreed with part of your comments and the amendments are listed below:
We have added additional information about the meeting and no financial compensations in the method section (lines 91 & 97).
There is a word limit in this paper therefore we could not include additional information regarding the stakeholder. However, in the acknowledgement section, we have already listed all stakeholders including their names and departments/sectors. No stakeholder is anonymised in this work and hence it is clear who was involved. However, we have added some text around the method section to make this clear how individuals approached including from a European network.
We strongly disagreed to quantify the experience that the stakeholder has. The meaning of the number is vague. However, since we have completed eligibility checks for each recruited stakeholder before the meeting conducted to ensure they have the relevant experience and knowledge, this has been added in the method section.
In the acknowledgement section, we have already mentioned which county the stakeholders are from, not only from Wales/UK. Additionally, a few stakeholders were recruited from a European network (they are either not in the UK or with experience collaborating with non-UK countries) and hence we don’t think there are any biases and therefore we don’t see the need on discussing potential biases in the paper.

Response 3: Thanks for spotting this error. This sentence has been removed.

Response 4: We have changed this sentence to “Funding for clinical trial methodology research in this area was thought to help improve the quality of clinical research findings”.

Response 4: We did not mention “ASIs are not funded” in the paper. Trials for ASIs are funded but the funding is not enough for doing the methodological work i.e. engaging in high-dimensional microbiological analysis. We have amended the text for this sentence to make this point clear - “However, given the lack of funding for methodological work in ASIs, there is insufficient time or capacity for statisticians to engage in high-dimensional microbiological analysis, and hence microbiological data are dichotomised or categorised during the analysis even though this oversimplifies the data and may not be appropriate”
For the point about dichotomised microbiological outcomes, we have already provided examples at the start of section 3.3 Perspectives on the current microbiological outcome.

Response 5: We have now added on “and involving in-depth discussion and understanding among microbiology, clinical, and statistical experts” after this sentence to make this clear what we meat. 

Response 6: Thanks for this. The second sentence is now removed.

Response 7: Funders and clinical trial researchers typically consider microbiological outcomes as less important than clinical outcomes. We have now amended this sentence to make this clear that there is a lack of communication during the trial decision - “Another critical fact point highlighted by the microbiologists was that a lack of sufficient communication between microbiologists and infectious disease doctors when designing and conducting these trials meant that microbiological outcomes were considered less important than other outcomes (e.g. clinical outcomes). Therefore, multiple separate views on the importance of microbiological outcomes in trials were found.”

Response 8: Thank you for this. However, we decided not to remove this paragraph. It is because this paragraph is about patient views. Any intervention aims to benefit the patient and we think what’s mattered to the patient is also important.

Response 9: Sorry, we don’t understand this comment.

Response 10: Thanks for this. We think the term “intermediate outcome” is well-known enough for trialists. We have included a reference about this, which has been published 10 years ago. But we don’t think there is a need for extending the sentence.

Response 11: This section is about how statisticians can do better for microbiological data. We agreed that compromises are often required between what is the best thing that can be done and what can be done. However, all stakeholders agreed that statistical methods also have value to improve the quality of the outcome, and arguably the dichotomisation of complex microbiological data does not extract the full value from these datasets. If the method is not improve, this means regardless of the quality of the data collection, the results provided by the current statistical methods did not provide very meaningful values. Hope you agree with us.

Response 12: Thank you for this, we decided not to merge points 1 and 2 but we agreed that these two points are similar. We have now made point 2 a subset of point 1.

Response 13: Database development should help provide a standardised format for data provided to a statistician so that they can make more efficient use of their time and focus on the application of appropriate statistical methods rather than data cleaning / wrangling.

Response 14: Thanks for this, we have now changed some wording around the conclusion section. Including extending what we mean by evidence.

Round 2

Reviewer 2 Report

I am satisfied with the reply and changes of the authors.